# Fast Adhesion of Gold Nanoparticles (AuNPs) to a Surface Using Starch Hydrogels for Characterization of Biomolecules in Biosensor Applications

**DOI:** 10.3390/bios10080099

**Published:** 2020-08-14

**Authors:** Frances L. Heredia, Pedro J. Resto, Elsie I. Parés-Matos

**Affiliations:** 1Department of Chemistry, University of Puerto Rico at Mayagüez, Mayagüez, PR 00680, USA; frances.heredia@upr.edu; 2Department of Mechanical Engineering, University of Puerto Rico at Mayagüez, Mayagüez, PR 00680, USA; pedroj.resto@upr.edu

**Keywords:** gold nanoparticles, starch hydrogels, immobilization, depletion stabilization, cell-free DNA biosensor

## Abstract

Gold nanoparticles (AuNPs) are the most thoroughly studied nanoparticles because of their remarkable optical properties. Color changes in assays that use AuNPs can be easily observed with the naked eye, resulting in sensitive colorimetric methods, useful for detecting a variety of biological molecules. However, while AuNPs represent an excellent nano-platform for developing analytical methods for biosensing, there are still challenges that must be overcome before colloidal AuNPs formulation can be successfully translated into practical applications. One of those challenges is the ability to immobilize AuNPs in a solid support. There are many difficulties with controlling both the cluster size and the adhesion of the coatings formed. In addition, many of the techniques employed are expensive and time-consuming, or require special equipment. Thus, a simple and inexpensive method that only requires common lab equipment for immobilizing AuNPs on a surface using Starch Hydrogels has been developed. Starch hydrogels confer a 400% increase in stability to the nanoparticles when exposed to changes in the environment while also allowing for macromolecules to interact with the AuNPs surface. Several starch derivatives were tested, including, dextrin, beta-cyclodextrin and maltodextrin, being dextrin the one that conferred the highest stability. As a proof-of-concept, a SlipChip microfluidic sensor scheme was developed to measure the concentration of DNA in a sample. The detection limit of our biosensor was found to be 25 ng/mL and 75 ng/mL for instrument and naked eye detection, respectively.

## 1. Introduction

Gold nanoparticles (AuNPs) are the most thoroughly studied nanoparticles because of their remarkable optical properties used in a variety of applications such as sensing, detection, and imaging, making them an excellent material for diagnostic purposes [1,2,3,4]. The basis of AuNPs optical properties is the coherent light absorption due to the coherent oscillation of the electrons in the conduction band caused by the electromagnetic field interactions [5,6,7]. This effect is generally referred to as Surface Plasmon Resonance (SPR), and in the individual atoms and bulk shape, this effect doesn’t take place [8,9]. The strong, vibrant color of the AuNP colloidal solution is caused by the SPR absorption [10]. The optical properties of AuNPs have been used in sensing biological molecules and cells because of the sensitivity and selectivity response to the environment [11,12,13]. Specific formulations of AuNPs have been manufactured previously by various research groups to target biological molecules, such as DNA, RNA, proteins, as well as cells, metal ions, small organic compounds, among others [14,15,16,17]. However, while AuNPs represent an excellent nano-platform for developing analytical methods for biosensing, there are still challenges that must be overcome before the fabricated colloidal AuNPs formulation can be successfully translated into clinical trial research. One of those challenges is the ability to immobilize AuNPs in a solid support. Chemical Vapor Deposition (CVD) is a process commonly used by the industry for preparing immobilized AuNPs on various substrates [18]. However, this method requires expensive equipment and has limitations in the formation of clusters on surfaces. Other physical methods, based on temperature decomposition of precursors, often led to variations in particle size. The deposition-precipitation (DP) method is also widely used for preparing catalytic powder materials with AuNPs. Through this technique, it is possible to obtain small gold metal particles with average of 1–3 nm in size on silica, Zirconia, titanium, and alumina supports [19]. Self-assembled monolayers (SAM) is another method to attach AuNPs to a surface. Those monolayers derived from trifluoroacetyl (TFA)-protected alkenylthiol, once assembled on a silicon surface, can bind nanoparticles at the distal end of the monolayer [20]. The electron transfer at these NP/insulator/metal structures is about 12 orders of magnitude more efficient than electron transfer between a metal and redox species in solution [21]. One should keep in mind that of the above-mentioned methods require additives such as thiols, strong acids (HF or Pirahna solution) and other stabilizers that are somewhat toxic. Another problem with AuNPs immobilization on the surface is the removal of organic molecules (i.e., surfactants) from the layer of gold nanoparticles [22]. In general, such additives are removed by heat treatment that sometimes it causes an agglomeration of nanoparticles, which makes difficult to control both the cluster size and adhesion of the formed coatings. Another strategy used to immobilize AuNPs is by integrating these particles into a hydrogel. Hydrogels have a broad range of properties, such as absorption potential, swelling ability, stability and degradation, bioadhesion and bioactivity, permeability, optical and surface mechanical properties [23,24,25,26], making them attractive materials for a variety of uses. Natural polymers are often used for synthesizing their composites with metals because of the non-toxic and biocompatible nature. Hydrogels are highly suited as promising materials for various applications, especially for diagnostic and therapeutic applications in biomedical areas because studies have been shown that can serve as scaffolds or adhesives between the surfaces of material and tissue [27,28].

Nanocomposite hydrogels can be generally synthesized through three different mechanisms that involve a combination of nanoparticles and hydrogels [29]. Among them include (1) AuNPs-hydrogel composites where Au (III) ions are integrated in poly(N-isopropylacrylamide) (PNIPAM) or poly(acrylic acid) (PAA) hydrogel networks, and these gold-functionalized polymer hydrogels are subsequently reduced or hydrolyzed [30]; (2) AuNPs-acrylamide hydrogel composites where AuNPs are added to an acrylamide monomer solution, and then a gelation catalyst is added to start polymerization [31]. This method has some disadvantages that hinder its wide-ranging applications including aggregation of nanoparticles in a monomer solution before and during the gelation process, and of leaching nanoparticles out of the hydrogel matrix if the cross link density is low [32]; and (3) AuNPs-hydrogel composites developed through an innovative approach by applying crosslinker groups on the surface of modified AuNPs to form the hydrogel matrix. Bond formation between nanoparticles and hydrogels is achieved via EDC coupling [33]. Overall, many of these methods are expensive, time consuming and require special equipment [20]. To overcome these hurdles, a simple, inexpensive, non-toxic and fast method for immobilizing AuNPs on a surface has been developed using four polysaccharide hydrogels and common laboratory equipment. Our new method can be used with preformed AuNPs, without the requirement of modifying their surface.

## 2. Materials and Methods

### 2.1. Chemicals

Gold nanoparticles (AuNPs) with a diameter of 20 nm (MW 196.97 g/mol; CAS 741965), Sigmacote siliconizing reagent (CAS SL2), Potato starch (CID 24836924; CAS 9005-25-8), Maltodextrin (CID 439586; CAS 9050-36-6), NaCl and MgCl_2_ were purchased at Sigma-Aldrich. Dextrin (CID 62698; CAS 9004-53-9) and beta-cyclodextrin (CID 444041; CAS 7585-39-9) were purchased from Acros Organics. Sheared Salmon Sperm DNA (CAS AM9680) and Invitrogen DEPC-Treated Water (CAS AM9906) were obtained from ThermoFisher Scientific. 

### 2.2. Gold Nanoparticles Starch Suspension

An aliquot (V = 1.5 mL) of AuNPs suspension was placed in a 2-mL Eppendorf tube (Eppendorf 4515D) and centrifuged at 14,000 rpm for 20 min. The supernatant (mostly citrate buffer) was removed until 40 μL of the AuNPs suspension have left. A second aliquot of AuNPs suspension was added and centrifuged again under the same conditions. The supernatant was removed until 50 μL of the AuNPs suspension have left. A stock solution of 0.1 g/mL Potato Starch was prepared by mixing 1.0 g of starch with 10.0 mL of deionized water. AuNPs-Starch suspensions were prepared by adding the appropriate amount of the starch stock solution to the concentrated AuNPs until the 1-mL mark of the tube. Tubes were heated in boiling water for 3 min while gently rocking side to side. Then, the tubes were let to cool down at room temperature for 1 h, at which time the solution had gelled. 

### 2.3. Surface Stability Test 

AuNPs-Starch suspensions were prepared at five different starch concentrations (50 mg/mL, 25 mg/mL, 10 mg/mL, 5 mg/mL, 2.5 mg/mL and 0 mg/mL). For each, drops of 10 μL were placed on a Hammerhill hydrophilic paper (Erie, PA, USA) and allowed to dry overnight at 20 °C. Images of each starch concentration on paper were acquired using an HP DeskJet Plus scanner (Hewlett-Packard, Palo Alto, CA, USA) and processed using ImageJ by converting images into gray scale and calculating the intensity with the Analyze tool. Results were then graphed in Microsoft Excel (Microsoft Office 365). All results were expressed as the mean ± SD.

### 2.4. UV/Vis Absorption Experiments

For the UV/Vis absorption assays, Corning 96-Well Plates (Corning, NY, USA) were used. In each well, 10 μL of the AuNPs suspension were added followed by 10 μL of a salt solution. The absorption spectra were recorded on a Synergy HTX spectrophotometer (BioTek; Winooski, VT, USA) from the Hauptman Woodward Institute, Buffalo, NY, USA. The wavelength was measured from 300 nm to 700 nm. Data processing analysis was done on Microsoft Excel (Microsoft Office 365). All results were expressed as the mean ± SD.

### 2.5. Microfluidic Device Development

A microfluidic device is developed following the Slip Chip approach [34]. The device is composed of three cast acrylic layers (McMaster-Carr product; Elmhurst, IL, USA), each having a thickness of 3.175 mm. The area of the device is 5 cm by 2.5 cm. The inlet and outlet ports are cut through the top layer and are 1.5 mm in diameter. The center layer has a microfluidic channel 40 mm in length, 1.5 mm in width and 3.175 mm in height. The bottom layer has an engraved microfluidic channel that matches that of the middle layer and is 40 mm in length, 1.5 mm in width and approximately 1 mm in depth. The three acrylic layers are placed on top of each other and held together by a press clamp. A white piece of paper is placed below the three acrylic layers in order to aid visualize colorimetric assay changes. The acrylic device was made using an EPILOG Mini CO_2_ 60 W laser cutter. Each layer is made individually from cast acrylic. The inlet and outlet ports in the top layer and the microfluidic channel in the middle layer are made by cutting all the way through their respective acrylic layers using the following settings (speed: 25%, power: 75%, frequency: 2500 Hz). The bottom layer has an engraved channel made by using the following settings (speed: 80%, power: 30%, frequency: 2500 Hz).

### 2.6. Biosensor Assay 

DNA stock solutions were prepared by heating an aliquot of Sheared Salmon Sperm DNA in boiling water for 5 min and quickly placing it on ice for 15 min. Serial dilutions from this aliquot were done to prepare eight DNA solutions with concentrations ranging from 10 ng/μL to 10 μg/μL. For biosensor assays, the bottom channel of the microfluidic device was filled with 20 μL of AuNP-Starch suspension at a starch final concentration of 10 mg/mL. The middle channel was filled with 4 M NaCl solution. The DNA solution was injected through the inlet into the bottom channel. The Slip Chip was incubated at room temperature for 3 min at which time the middle layer would be switched so the NaCl solution interacted with the DNA-AuNPs mixture. A change in color on the solution was detected by the naked eye as well as UV/Vis characterization.

## 3. Results and Discussion

### 3.1. Gold Nanoparticles Fixed to a Surface using Starch Hydrogels

To test the ability of Starch hydrogels to fix AuNPs to a surface, five solutions of Starch-AuNPs were prepared at different concentrations. Drops of these solutions were placed on paper and allowed to air dry. After the drops were fully dry, the color difference between the adsorbed samples was discerned visually and it was noticeable that those samples with the highest concentration of Starch allowed AuNPs to remain stable under dry conditions, this is known because of the red color of the sample indicating that the AuNP is not aggregated, while those samples with low or no Starch caused AuNPs to aggregate upon drying (Figure 1A). Papers were scanned to quantitatively confirm color differences by turning their color images into grayscale, and calculating the average intensity. As shown in Figure 1B, there is no difference between the control and the 2.5 mg/mL Starch solution. However, there is a small difference between the control and the 5 mg/mL Starch solution, which could be that the AuNPs are partially aggregated, as can be observed by the slight change in color from red to purple.

### 3.2. Quantitation of the Effect of the Starch-Hydrogel on the Gold Nanoparticle

Gold nanoparticles and Starch-suspended gold nanoparticle samples were analyzed using a UV-Vis spectrophotometer. Figure 2 shows the spectra of different AuNP samples before and after the addition of 2 M NaCl solution. Generally, the presence of NaCl reduces the repulsive interactions between these gold nanoparticles, and causes its aggregation. Suspending gold nanoparticles in a Starch hydrogel seems to stabilize the particles against aggregation, therefore, there is no color change when a salt is added to the solution. Thus, solutions with adequate amounts of Starch prevents the aggregation process and can remain red in color, whereas in solutions without Starch, AuNPs can aggregate after addition of salt, and a change in color to blue was observed. These results suggest that the addition of Starch can stabilize AuNPs through depletion stabilization [35,36,37], and in the absence of Starch, AuNPs are stabilized by charge repulsion [38]. Dispersed AuNPs show a small absorption peak at 520 nm (Figure 2, blue curve) due to a surface plasmon phenomena [39]. Meanwhile aggregation of AuNPs causes a red shift of the small peak to 630 nm (Figure 2, red curve). Gold nanoparticles suspended in Starch have an increase in absorbance intensity as well as a slight red shift to 525 nm, consistent with a depletion stabilization effect that has been found when polymers interact with gold nanoparticles [35,36].

A quantitative study using UV-Vis absorption spectroscopy was performed to analyze the effect of depletion stabilization. The peak ratio at 630 nm and 525 nm was employed to quantitatively scale the aggregation effect on the particles since there is a linear dependence of the Abs_630_/Abs_525_ ratio level of aggregation [40]. Therefore, the aggregation state of gold nanoparticles can be quantified, and a higher ratio indicates aggregated gold nanoparticles. To study the stability effect of Starch on the gold nanoparticles, three AuNPs-Starch samples were titrated with increasing amounts of NaCl solution, as shown in Figure 3A. In the absence of Starch (light blue), AuNPs began to aggregate upon addition of 25 mM NaCl and became fully aggregated at 1 M NaCl. Meanwhile, the 10 mg/mL of Starch-AuNPs suspension began to aggregate at 100 mM NaCl (yellow curve) and became fully aggregated at 1 M NaCl. Those AuNPs suspensions containing 25 mg/mL and above of Starch effectively stabilized the gold nanoparticles (dark blue and green curves).

Starch-AuNPs suspensions were also titrated with increasing amounts of MgCl_2_ to test the limit of stability that Starch provides (Figure 3B). Magnesium chloride salts were used due to its high ionic strength. Aggregation of AuNPs started at a concentration of 25 mM MgCl_2_ when the concentration of Starch is less than 10 mg/mL, reaching full aggregation at 500 mM MgCl_2_. But when AuNPs were suspended in 10 mg/mL of Starch, it became fully aggregated at 1 M MgCl_2_, in 25 mg/mL of Starch, it became fully aggregated at 4 M MgCl_2_ and in 50 mg/mL of Starch, the particles remain stable up to 100 mM MgCl_2_ and it never became fully aggregated.

This observed level of stabilization against both salts is quite remarkable, especially when taking into account that AuNPs were not covalently modified. The mechanism by which Starch can stabilize AuNPs could be explained as either depletion stabilization or steric stabilization. In depletion stabilization, the polymer concentration is high enough for the repulsive energy barrier to allow the particles to be kinetically stabilized [37], meanwhile in steric stabilization the particle surface is coated with polymers, and the size of the polymer is greater than the London interaction range and blocks the particle surface from absorbing other molecules [35].

### 3.3. Determination of the Stabilization Mechanism

For many applications involving nanoparticles, solving the colloidal stability problem while still retaining surface accessibility is highly desirable. Steric stabilization, however, cannot achieve this goal. In order to distinguish which mechanism was taking place, DNA solutions were added to different AuNP-hydrogel suspensions to see if they could bind to the AuNP. It is known that negative charge on the DNA backbone allows the DNA strand to adsorb onto the nanoparticle and stabilizes the AuNP against aggregation at concentrations of salt that would ordinarily cause aggregation [41]. The addition of the DNA solutions further improved the stability of the AuNP-hydrogel suspensions, indicating that the method of stabilization was depletion and not steric (data not shown).

### 3.4. Development of a Microfluidic Device for the Detection of Cell-Free DNA

Capitalizing on the ability of the AuNP-suspension to absorb other molecules, we decided to test if the AuNP fixed with the starch hydrogels could be used as a substrate to immobilize the AuNP in a biosensor. A Slip Chip sensor scheme (Figure 4A) was developed where gold nanoparticles are fixed to the bottom of a microfluidic channel using the starch hydrogels, meanwhile there is a reservoir of oversaturated NaCl solution in the top channel. When the Slip Chip is open (Figure 4B, left image), the DNA sample can be injected into the bottom channel, but in closed position (Figure 4B, right image), the salt solution interacts with the AuNP-DNA suspension. Surface stabilized AuNPs could be used as a method to develop biosensors. Several research groups have proposed that if ssDNA and dsDNA with sticky ends are exposed to AuNPs, its helical chains became sufficiently flexible to partially uncoil [42,43] and, under these conditions, attractive van der Waals forces emerge between bases and gold nanoparticles, thus allowing DNA to bind closer to the AuNPs surface [41]. After AuNPs are incubated with the DNA solution, the red color of the AuNPs remains at a high enough DNA concentration, meanwhile it turns to blue at low DNA concentration. To further characterize the detection range of this assay, a series of samples containing different concentrations of DNA have been tested on the Slip Chip (Figure 5), and subsequently, placed in the spectrophotometer to measure the absorbance at 630 nm and 525 nm. The resulting calibration plot of Abs_630_/Abs_525_ ratio versus the known concentration of DNA was plotted (see Figure 6). There is a linear correlation between the Abs_630_/Abs_525_ ratio and the DNA concentration in the range of 0–10 μg/mL. The detection limit of this biosensor was found to be 25 ng/mL for instrument detection and 75 ng/mL for detection with the naked eye. This detection limit is lower than previously reported for AuNPs immobilized with the SAM method, which has a detection limit of 0.58 μg/mL [44].

This is a proof-of-concept device for the detection of circulating-free DNA (cfDNA), since cfDNA in plasma is associated with cancer and other pathologies. Circulating DNA is present in the blood of all individuals, but it has been found that patients with cancer have far greater levels of circulating DNA in their blood than do healthy subjects [45]. However, cfDNA is not limited to the serum or plasma of cancer patients, as large levels of cfDNA can also be found in patients with other pathological conditions such as systemic lupus erythematosus, rheumatoid arthritis, glomerulonephritis, pancreatitis, cholelithiasis, inflammatory bowel disease, peptic ulcer disease, hepatitis, esophagitis, pulmonary embolism, ulcerative colitis, and Miliary tuberculosis [46], which are associated with inflammatory processes. As shown in Table 1, the detection limit of our biosensor is sensitive enough to discriminate between a healthy individual and an individual with pathological levels of cfDNA.

In addition to studying the effect that Starch hydrogels may have on the stability of gold nanoparticles, other starch derivatives were also studied including dextrin, maltodextrin and *β*-cyclodextrin. Dextrin is a mixture of polymers of D-glucose units linked by *α*-(1→4) and *α*-(1→6) glycosidic bonds produced by the hydrolysis of Starch or Glycogen [49]. Maltodextrins are oligomers of D-glucose, primarily linked with *α* (1→4) glycosidic bonds [50] and *β*-cyclodextrin is a cyclic derivative of starch, prepared from partially hydrolyzed starch (maltodextrin) by an enzymatic process. All three polysaccharides are able to form hydrogels [51].

To study the stability effect of more hydrogels on the gold nanoparticles, other AuNPs-Starch samples were titrated with increasing amounts of NaCl solution (Figure 7). In the absence of these polysaccharides, AuNPs began to aggregate upon addition of 50 mM NaCl and became fully aggregated at 1 M NaCl. Meanwhile, Dextrin suspension did not let to aggregate at all, instead Dextrin completely stabilized AuNPs from the presence of ionic forces. Other two polysaccharide hydrogels did not show the same stabilization effect. For *β*-Cyclodextrin, these samples began to aggregate at 50 mM of NaCl and reached full aggregation at 200 mM, while the control samples reached full aggregation at 1 M. Meanwhile with Maltodextrin an opposite effect was observed because the AuNPs got aggregated even faster, at 25 mM (versus the 50 mM of the control sample) and, in the case of the 10 mg/mL suspension, samples became fully aggregated at 100 mM of NaCl. It is also important to note that the 25 mg/mL and the 50 mg/mL solutions of Maltodextrin/AuNPs suspension could not be prepared because the solutions aggregated upon contact with the AuNPs. When comparing the structures of these four polysaccharides, it can be seen that Starch and Dextrin have in common the *α*-1,6-glycosidic bond, allowing to form a branched and more complex structure. It is possible that these branches are involved in providing stability to the AuNP, but more research is needed to understand the exact mechanism of action.

## 4. Conclusions

A method was developed by which AuNPs can be bound to a surface. The Starch hydrogel suspension has shown that it can stabilize the AuNPs by depletion stabilization, where at a certain polymer concentration, the repulsive energy barrier becomes high enough to allow particles to be kinetically stabilized. A depletion stabilization instead of steric stabilization was considered herein because the Starch hydrogel should block the particle surface from absorbing other molecules. The experiment with DNA solutions also suggests that other molecules are still able to interact with these AuNPs. Additionally, a proof-of-concept of a label-free, colorimetric biosensor that utilizes immobilized AuNPs on an optically transparent substrate was developed. This biosensor provides a straightforward and efficient system that can be easily implemented in most laboratories. While the findings herein are promising, a lot still remains to be done to optimize this device for clinical trial research. These include: (1) further characterization of the biosensor’s sensitivity and dynamic range as a function of the size of the DNA fragment and its binding affinity in absolute terms, i.e., molecules per unit area, (2) direct kinetics and sensitivity in steady-state reactions, and (3) characterization in terms of the effect of colloid size and shape on the biosensors respon

## Figures and Tables

**Figure 1 biosensors-10-00099-f001:**
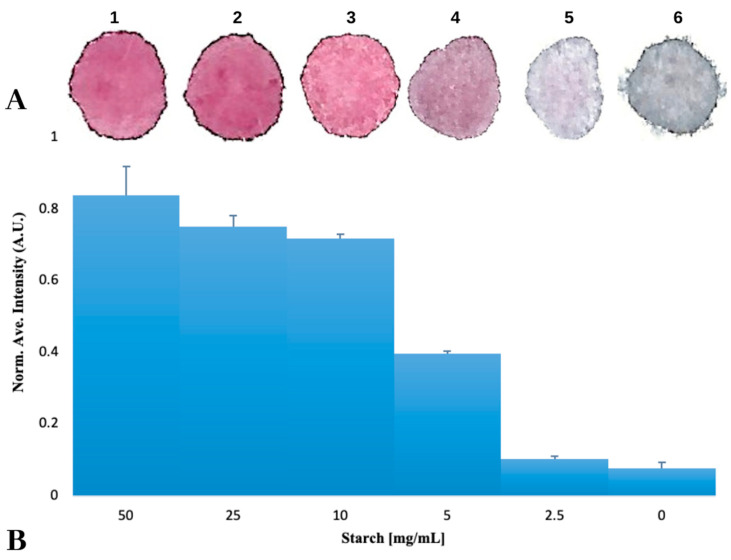
Normalized average intensity of the dry AuNP samples with different starch concentrations, fixed on a paper surface. (**A**) Scanned images of AuNPs on paper with different concentrations of Starch. (**B**) A histogram of the average intensity of each sample. Error bars represent the standard deviation of the intensity distribution of the sample.

**Figure 2 biosensors-10-00099-f002:**
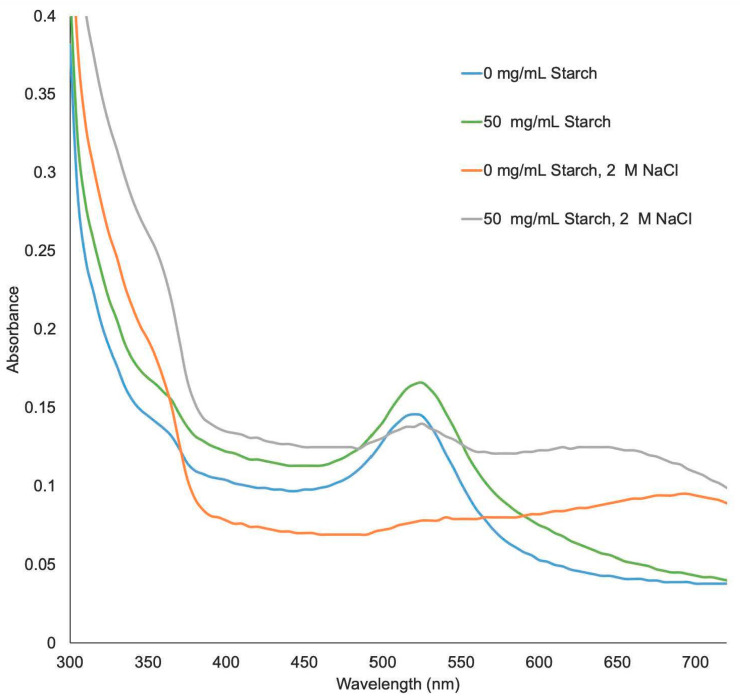
UV−Vis absorption spectra of AuNPs with and without Starch and/or NaCl.

**Figure 3 biosensors-10-00099-f003:**
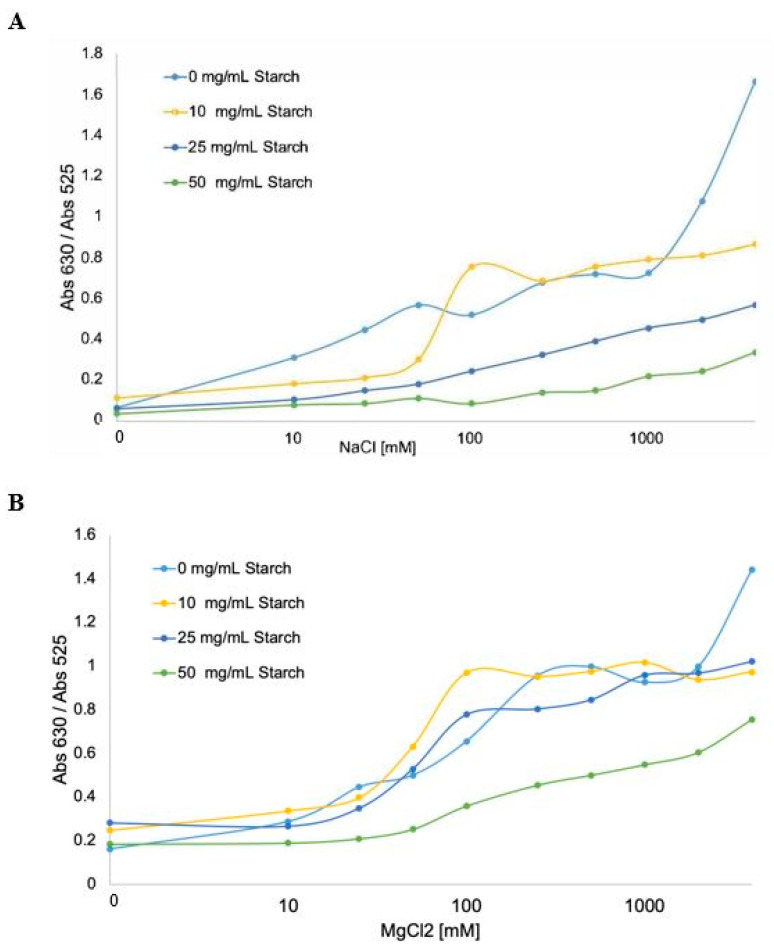
Plots of the concentration of salt versus absorbance ratio (Abs_630_/Abs_525_). The effect of different concentrations of salt on the aggregation level of AuNPs at different concentration of Starch solution is shown. (**A**) NaCl; (**B**) MgCl_2_.

**Figure 4 biosensors-10-00099-f004:**
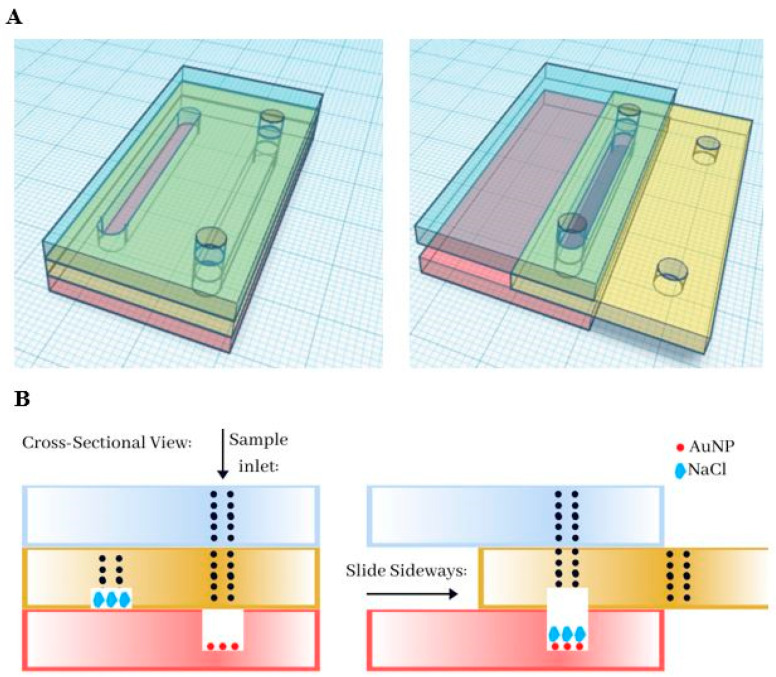
A Slip Chip manipulation schematic and its operating slots for fluidic function. (**A**) Side view of the Slip Chip assembly before and after salt addition. Top layer (in blue) is the sample inlet and the railing to keep bottom layers in place; Middle layer (in yellow) sample inlet flow line connecting to the bottom layer. The middle channel is prefilled with an oversaturated solution of NaCl. Bottom layer (in red) contains another channel, prefilled with AuNPs that were fixed to the surface using starch hydrogels. (**B**) When all three layers are in place and aligned, the sample is injected through the Sample Inlet, and then, incubated with the nanoparticles for 3 min. After the incubation step, the middle layer slides sideways to allow the salt reservoir to get in contact with AuNPs and the sample.

**Figure 5 biosensors-10-00099-f005:**
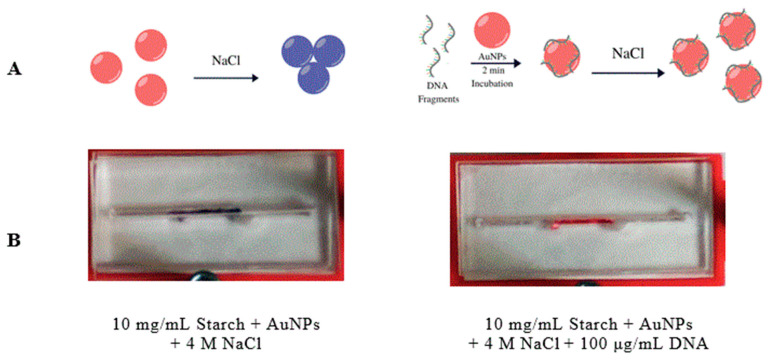
A scheme of the reactions that take place in the biosensor. (**A**) AuNPs exposed to an oversaturated salt solution (NaCl) in the absence of DNA (left) can aggregate and its color changes to blue. AuNPs incubated with DNA prior to salt addition (right), do not aggregate and the red color remains. (**B**) Images of the biosensor were taken showing colorimetric responses.

**Figure 6 biosensors-10-00099-f006:**
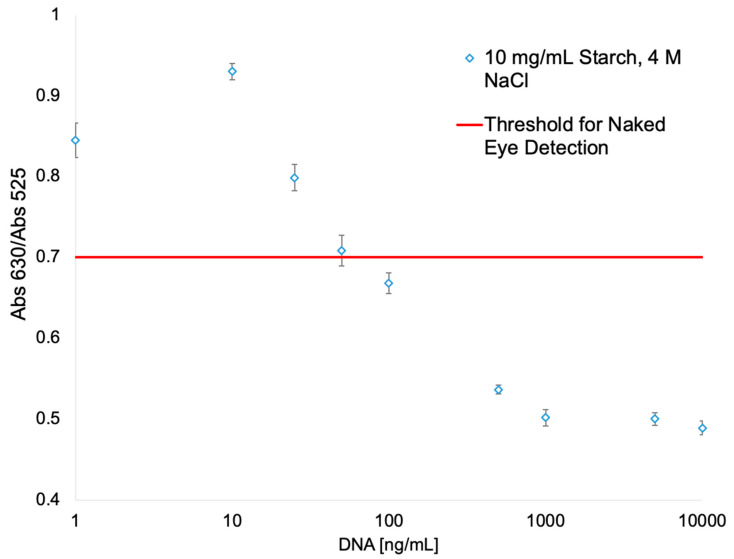
Plot of DNA concentration versus absorbance ratio (Abs_630_/Abs_525_) for the biosensor assay. The red line indicates the threshold at which the naked eye can detect the change in color of the AuNPs solution.

**Figure 7 biosensors-10-00099-f007:**
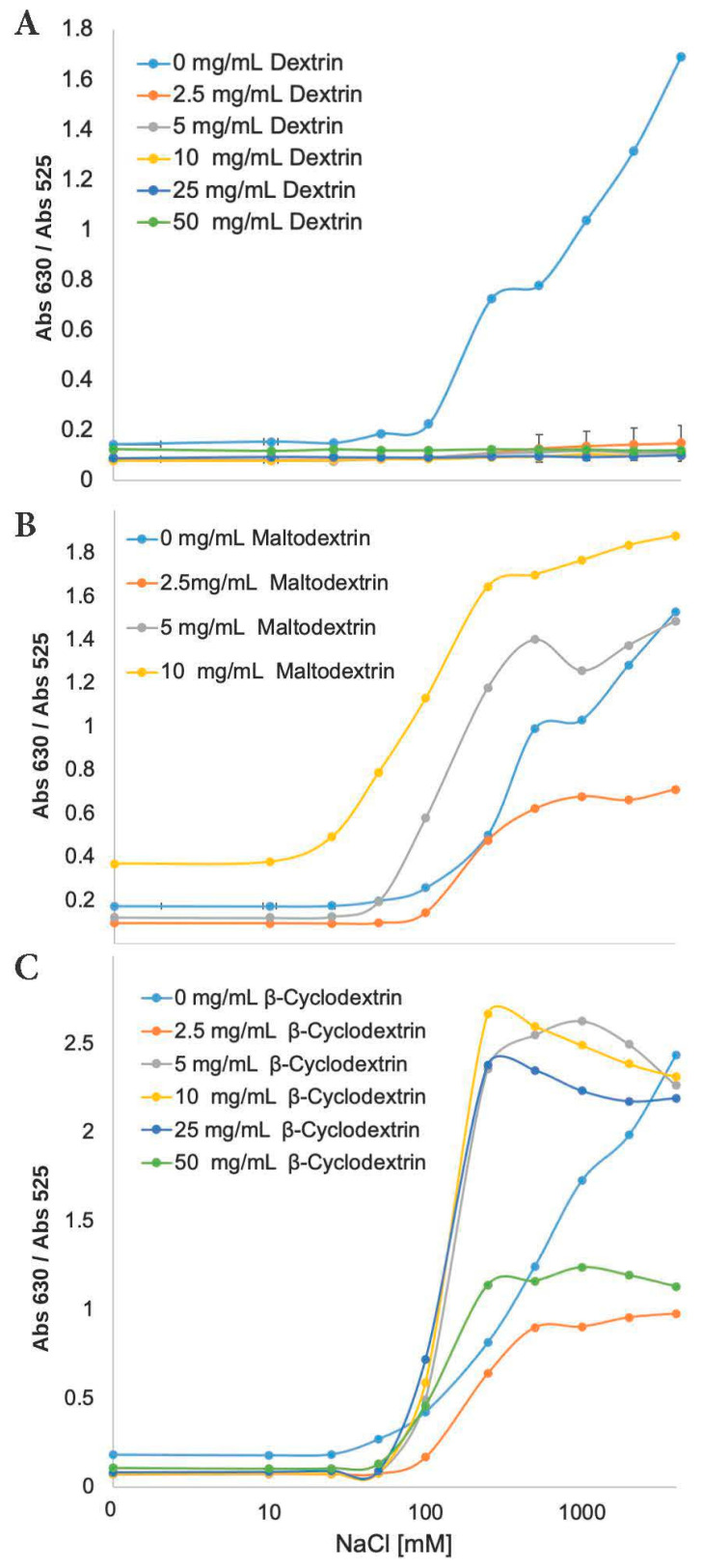
Plots of Abs_630_/Abs_525_ ratio of AuNPs in three different polysaccharide hydrogels suspension. (**A**) Dextrin, (**B**) Maltodextrin and (**C**) β-Cyclodextrin. Each sample was run at five different concentrations of polysaccharides versus known concentration of NaCl.

**Table 1 biosensors-10-00099-t001:** Summary of mean concentration of cfDNA on different types of cancer.

Pathology	cfDNA (ng/mL)	Reference
Healthy Control	20.52	[47]
Colorectal Cancer	70.32	[48]
Neck Cancer	65.50	
Breast Cancer	37.00	
Melanoma	40.00	
Sarcoma	49.00	
Brain Cancer	6.00	

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
