# Peer review of "Fast Adhesion of Gold Nanoparticles (AuNPs) to a Surface Using Starch Hydrogels for Characterization of Biomolecules in Biosensor Applications"

_biosensors, 2020, doi:10.3390/bios10080099_

Round 1
Reviewer 1 Report
This article by Pares-Matos and co-workers deals with the adhesion of gold nanoparticles on solids, and specifically it trys to address challenges in the immobilization of the particles on a solid substrate. To this end they explore the use of hydrogels, and study the stability of the construct. They also include a proof of concept application in which they sense DNA interactions and monitor these colorimetrically.
Definitely a nice paper, well written and well presented, and therefore I only have a few minor remarks to make:
- Early literature on immobilization and stability of gold nanoparticles on solid substrates, such as gold and silicon electrodes, is largely ignores. Few seminal papers should be mentioned in the introduction when introducing the analytical power of AuNPS and when listing the challenges still not met. For example, gold nanoparticles are great as optical markers, but equally great as charge-transfer entities. Adsorption of particles can be ensure using very simple chemistries, and as such the authors should acknowledge these options and achievements of others, citing for example:
- Le Saux, S. Ciampi, K. Gaus and J. J. Gooding, ACS Appl. Mater. Interfaces, 2009, 1, 2477-2483; J.-N. Chazalviel and P. Allongue, J. Am. Chem. Soc., 2011, 133, 762-764.
2) Figure 1. How are the intensities obtained and calibrated? At least I would normalize the intensity y-axis units.
3) Figure 3. Is it really necessary to take up a full article page for a diagram of a liquid cell? Supporting Information?
4) With regards to the colorimetric detection of DNA. Why the DNA adsorb on the particle? What is the net charge of the particle?
5) Figure 8. What is the value of adding chemical structures of known chemicals? Especially since the chemical details of these sugars are not referred to in the manuscript.
Reviewer 2 Report
This is a work about the low-cost immobilization of gold nanoparticles on a surface with the help of oligo-or poly-saccharide hydrogels. The authors perform some basic tests with a microfluidic bionsensor unit.
The studied subject is one of substantial interest, the work performed is systematic and reasonable (though of somewhat limited range) as regards choices.
Two key issues:
- Is there any other gel-based method forAuNP immobilization? The gel might be based on any appropriate macromolecular substance. If there is/are, it/they should be mentioned; if there is not, one should wonder why, as the concept is not an unusual one. Actually AuNP immobilization with gels is a relatively common practice ; the authors should refer clearly to pertinent works and indicate clearly what is common and what is different (e.g. the method might be similar but the application (bio-related or other, e.g. catalysis) different).
- The detection limits should be compared to those in the literature for other AuNP-immobilization methods.
Round 2
Reviewer 2 Report
The authors have responded satisfactorily to my points.